# Performance of offline passive tracer advection in ROMS (v3.6, revision 904)

Kristen M. Thyng[1], Daijiro Kobashi[1], Veronica Ruiz-Xomchuk[1], Lixin Qu[2], Xu Chen[3], and Robert D. Hetland[1]

[1]Department of Oceanography, Texas A&M University, College Station, TX, USA
[2]Department of Earth System Science, Stanford University, USA
[3]Center for Ocean-Atmospheric Prediction Studies, Florida State University, Tallahassee, FL, USA

**Correspondence:** Kristen Thyng (kthyng@tamu.edu)

**Abstract.** Offline advection schemes allow for low computational cost simulations using existing model output. This study presents the approach and assessment for passive offline tracer advection within the Regional Ocean Modeling System (ROMS). An advantage of running the code within ROMS itself is consistency in the numerics on and offline. We find that the offline tracer model is robust: after about 14 days of simulation (almost 60 units of time normalized by the advection timescale), the

skill score comparing offline output to the online simulation using the `TS_U3HADVECTION` and `TS_C4VADVECTION` (3rd-order upstream horizontal advection and 4th-order centered vertical advection) tracer advection schemes is 99.6% accurate for an offline time step 20 times larger than online, and online output saved with a period below the advection timescale. For tracer advection scheme `MPDATA`, accuracy is more variable with offline time step and forcing input frequency choices, but is still over 99% for many reasonable choices. Both schemes are conservative. Important factors for maintaining high offline accu-

racy are: outputting from the online simulation often enough to resolve the advection timescale, forcing offline using realistic vertical salinity diffusivity values from the online simulation, and using double precision to save results.

*Copyright statement.* TEXT

## 1 Introduction

The ability to integrate Eulerian tracer fields offline, or separate from the online, original full simulation is attractive because

of the improved computational efficiency. Once an online simulation has been run, any number of offline simulations can be run, forced by the stored online model output, using a larger time step, and only needing to integrate the transport field itself. This allows for many simulations when with the online simulation fewer would have been possible. This study presents the development and assessment of an offline passive tracer advection model that is part of the Regional Ocean Modeling System (ROMS), version 904, in COAWST (Shchepetkin and McWilliams, 2005; Warner et al., 2010). While the ultimate goal of

this work is to run ROMS with both offline floats representing oil and tracers representing biological processes, along with sediment–oil interactions, the present focus is on the offline tracer model with a passive tracer.

Previous work has been done in this area with other models. An offline tracer model for the MIT general circulation model (MITgcm) was developed and showed good accuracy (Hill et al., 2004). For a set output frequency, an offline time step of 8 times the online time step gave a skill score of over 98%. An offline tracer model based on MITgcm has been used in several studies (Dutkiewicz et al., 2001; McKinley et al., 2004). Another offline tracer model, OFFTRAC, is based on the Hallberg Isopycnal Model (HIM) and has been used for long-term biogeochemical integration (Zhang et al., 2014). Other tracer models have been developed separately from a full numerical ocean simulator. Gillibrand and Herzfeld (2016) have developed a separate tracer advection model that is not numerically limited by the Courant number as is expected in the present case. Another such model developed by Khatiwala et al. (2005) has a different approach entirely to offline tracer advection, using a mathematical approach that is distinct from more commonly used numerical tracer integration. Interestingly, Lévy et al. (2012) found that for particular dynamical scenarios, degrading online model output spatially can result in offline computational savings with little accuracy degradation.

The offline tracer model described in this paper is integrated into and derived from the ROMS model: preprocessor flag choices allow access to the offline capability. While not being derived from a specific ocean model allows for wider potential use, as in some of the previously-described models, there may be an advantage in using the offline model that is derivative of the offline model to ensure consistency, using the exact same numerics and setup. The expected user for this software is someone who uses ROMS for their ocean modeling needs and wants to have the ability to run more tracer simulations, decoupled from their more expensive online simulations. Another type of user may simply have some ROMS output available, and this code will allow them to leverage it beyond its originally intended use.

The experimental setup is described in Section 2; this includes the description of the model setup (Section 2.1), the offline experiments (Section 2.2), and the metrics used for evaluation (Section 2.3). Results are shown in Section 3, and a discussion of results is in Section 4. Specific code descriptions are in the Appendix: code changes made for robust offline tracer advection (Section B) and a description of how to set up both online and offline simulations for best results is given in Section C.

## 2 Experimental setup

### 2.1 Online model setup

The online model is set up for the northern Gulf of Mexico (25.6°N–30.6°N, 94°W–84°W). The domain was chosen because the final goal of this work is to simulate the fate of oil spilled in this region in 2010. The horizontal resolution is 0.04 degree to fully resolve mesoscale processes, and there are fifty vertical layers with refinement at the seabed and sea surface (transformation equation parameter `Vstretching` is 5 and stretching function parameter `Vtransform` is 2) (Azevedo Correia de Souza et al., 2015). The time step is 20 seconds. HYCOM Global Reanalysis data (experiment "GLBu0.08/expt_19.1") is used to initialize the model and provide boundary conditions (Fox et al., 2002; Cummings, 2005; Chassignet et al., 2007; Cummings and Smedstad, 2013). The surface forcing is provided by hourly Climate Forecast System Reanalysis (CFSR) data (Saha et al., 2010) and air-sea turbulent fluxes are calculated using bulk formula COARE 3.0 (Fairall et al., 2003). In order to realistically simulate the water properties and dynamics in the coastal area, 21 daily river discharges with specified water transport flux

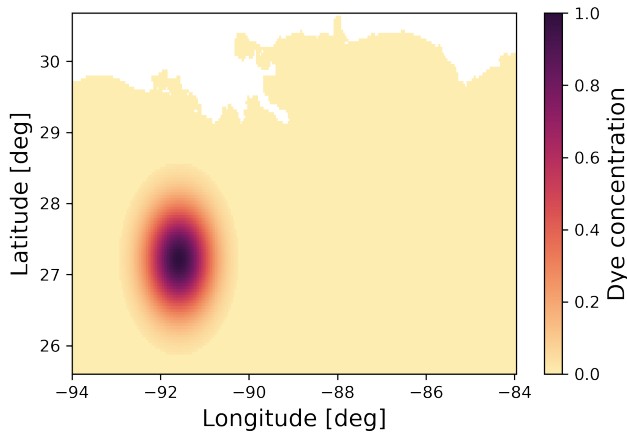

**Figure 1.** Model domain and initial blob of passive dye at the surface for the first set of numerical experiments.

and temperature (from USGS) are implemented as boundary fluxes along the coast. To stabilize the open boundaries, lateral nudging layers are set at the open ocean boundaries. The nudging time scale is 0.04 days at the boundary and gradually tuned to 10 days at the 18th interior grid. The climatology used for the nudging process is also provided by the HYCOM output. This online model ran for 90 days, from April 20 to July 19, 2010, but a subset of 14 days are used for the present experiments.

## 2.2 Offline experiments

### 2.2.1 Full water column Gaussian

A series of online and offline simulations were run to evaluate comparative performance of offline tracer advection. The first set of numerical experiments presented in this paper were initialized with a discrete Gaussian blob of dye southwest of the Mississippi river delta in a regional model of the north Gulf of Mexico (see Figure 1). The blob of dye extended fully through the water column. Online simulations were run with two tracer advection schemes: `MPDATA` (both horizontal and vertical) and `TS_U3HADVECTION` (3rd-order upstream horizontal advection) and `TS_C4VADVECTION` (4th-order centered vertical advection) (shortened to `U3C4` for the remainder of the paper). Additionally, the online simulations were output at different frequencies, as multiples of the time step (the `nhis` and `navg` parameters for ROMS):

$$\mathtt{nhis} = 1, 2, 5, 10, 20, 50, 100, 200, 500, 1000, 2000, 5000.$$

Given that the time step of the online simulation was 20s, these correspond to output frequencies of about 20s, 40s, 100s, 3.3min, 6.7min, 16.7min, 0.56h, 1.1h, 2.8h, 5.6h, 11h, and 28h. Online simulations were saved as both instantaneous snapshots (`his` files from ROMS) and as averages across time steps (`avg` files from ROMS).

These offline simulations were run using one of the two tracer advection schemes, with either `his` or `avg` files as climatology at the output frequency from the online simulations (controlled by `nhis`/`navg`). Additionally, they were optionally

forced by the vertical salinity diffusion variable, `Aks`, as calculated by the online simulation or by just the background value. Finally, for an input climatology file from an online simulation of a given output frequency (`nhis/navg`), offline simulations were run with a time step from the list of `nhis` values of up to the same output frequency as the online simulation. A time step of 50 times the online time step was found to lead to unstable solutions, so in effect, the offline time step could be 1, 2, 5, 10,

or 20 times the online time step, but never larger than the `nhis` value for the given simulation. Also note that the offline time step needs to divide evenly into the output frequency in the climatology file so that only two time steps are being accessed at a time. So, for a climatology forcing file of `nhis=50`, the offline simulation could not be forced with `dt=20`.

The relevant controlling timescale for this simulation is the advection timescale. Results from online simulations of the dye advection show a representative length scale of about $L = 10$km and speed of about $U = 0.5$m/s, giving an advection timescale

of $T = L/U = 20000$s, or about 5.6hr. This timescale is specific to the location of the dye patch, which is off the continental shelf and responding to mesoscale processes. If the dye patch was on the shelf, one would expect a shorter timescale. The timescale will be used to normalize times given in results and to interpret accuracy in relation to offline time choices.

### 2.2.2 At-depth realistic Gaussian

Another set of simulations was run to apply the lessons learned in the first set to a more realistic test case (Figure 2). This test

case is meant to represent an infusion of some material to the ocean at depth, for example dissolved methane gas. However, since in the present study we are testing only the passive offline tracer advection scheme, the tracer is passive, and has no particular behavior specific to a material. The dye is initialized in a discrete Gaussian blob at 800 m depth between 28 and 29°N latitude. Building off information from the previous simulations, only two offline simulations were run: one with online output frequency forced of `nhis=100` (about 30 minutes) to be a "good resolution" test case, and one with `nhis=1000` (about

5.5 hrs) to be a "low resolution" test case; both were run with the U3C4 tracer advection scheme and an offline time step of 20 times the online time step, or 400 seconds.

### 2.3 Metrics

### 2.3.1 Skill score

The main metric used to evaluate the performance of this model is a skill score, SS (Bogden et al., 1996; Hill et al., 2004;

Hetland, 2006). This is calculated as:

$$
\mathrm{SS} = 1 - \frac{\sqrt{\langle (D_\mathrm{on} - D_\mathrm{off})^2 \rangle}}{\sqrt{\langle D_\mathrm{on}^2 \rangle}}, \tag{1}
$$

where $D_\mathrm{off}$ and $D_\mathrm{on}$ are the volume of dye on the 3D grid and in time for the off and online simulations, and the brackets $\langle . \rangle$ indicate averaging over horizontal and vertical dimensions, returning a time series.

Often skill scores are calculated with respect to a reference. For example, for numerical model performance, the difference

between model and data in the numerator may be compared with the difference between climatology and data in the denom-

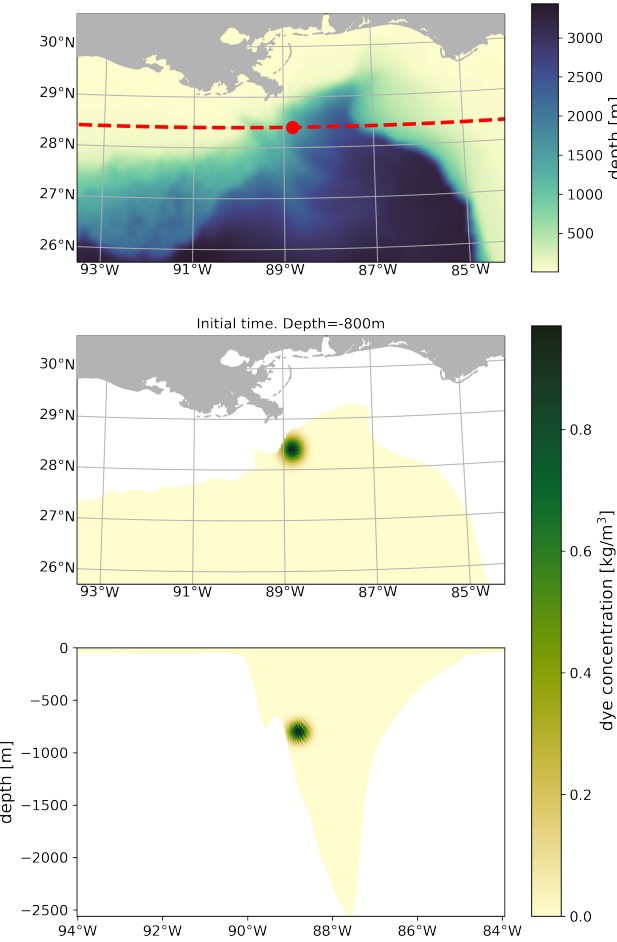

**Figure 2.** Second experiment: discrete Gaussian blob of dye at 800 m depth. Subplots show domain and bathymetry (top), dye slice at 800 m depth (middle), and vertical cross section of dye field (bottom) across the red dashed line in the top subplot. Red circle in top subplot indicates the center of the dye blob.

inator in order to assess how much better the model is performing than simple climatology (Hetland, 2006). An analogous comparison may be made here versus persistence of the initial condition of the dye patch, so that this skill score shows how well the offline model performs compared to simply persisting the initial condition:

$$\text{SS}_p = 1 - \frac{\sqrt{\langle (D_{\text{off}} - D_{\text{on}})^2 \rangle}}{\sqrt{\langle (D_{\text{initial}} - D_{\text{on}})^2 \rangle}}, \tag{2}$$

Skill scores are a comparison between an offline simulation and the online simulation from which it is forced, unless otherwise noted, so that the skill score represents accuracy of the offline simulation to the online simulation, or the skill in faithfully reproducing the online simulation. This is different from a measure of the accuracy of the online simulation itself to simulate the dynamics.

### 2.3.2   Percent error

Percent error is used to demonstrate the accuracy of the second set of simulations in space because it is not averaged over spatial dimensions like the skill score. The percent error at time $t_0$ is calculated as:

$$\text{E}(t) = \frac{|D_{\text{on}}(t_0, z, y, x) - D_{\text{off}}(t_0, z, y, x)|}{V_{\text{on}}(t_0, z, y, x) d_{max}(t_0)}, \tag{3}$$

where $D_*(t_0, z, y, x)$ is the on or offline dye volume at time $t_0$ in space (kg), $V_{\text{on}}(t_0, z, y, x)$ is the online volume of the grid cells (m$^3$), and $d_{max}(t_0)$ is the maximum dye concentration at $t_0$ (kg/m$^3$). The percent error represents the difference in the
offline from the online simulation compared to the maximum possible dye mass at that time step.

### 2.4   Simulations and software

Simulations were performed on a Linux cluster with 84 processors for online simulations and 28 for offline. The number used was not optimized. Analysis was performed in a Jupyter notebook (Kluyver et al., 2016) using pandas (Wes McKinney, 2010), xarray (Hoyer and Hamman, 2017), and scipy (Virtanen et al., 2020) for analysis, and Matplotlib (Hunter, 2007) for figures
with cmocean (Thyng et al., 2016) for colormaps.

## 3   Results

### 3.1   Full water column Gaussian

The accuracy of selected offline simulations are presented here. Since there were over 300 offline simulations, only selected results are shown to best illustrate specific points and show the overall performance of the model under a range of parameter
choices. Offline simulations are forced by snapshots of online output (`his`, not `avg` files) in all cases unless specified. These results are specific to this model setup and the dynamics that are being captured in the region, but should give specific results for other geographically-interested users with similar model setups and general guiding results for others.

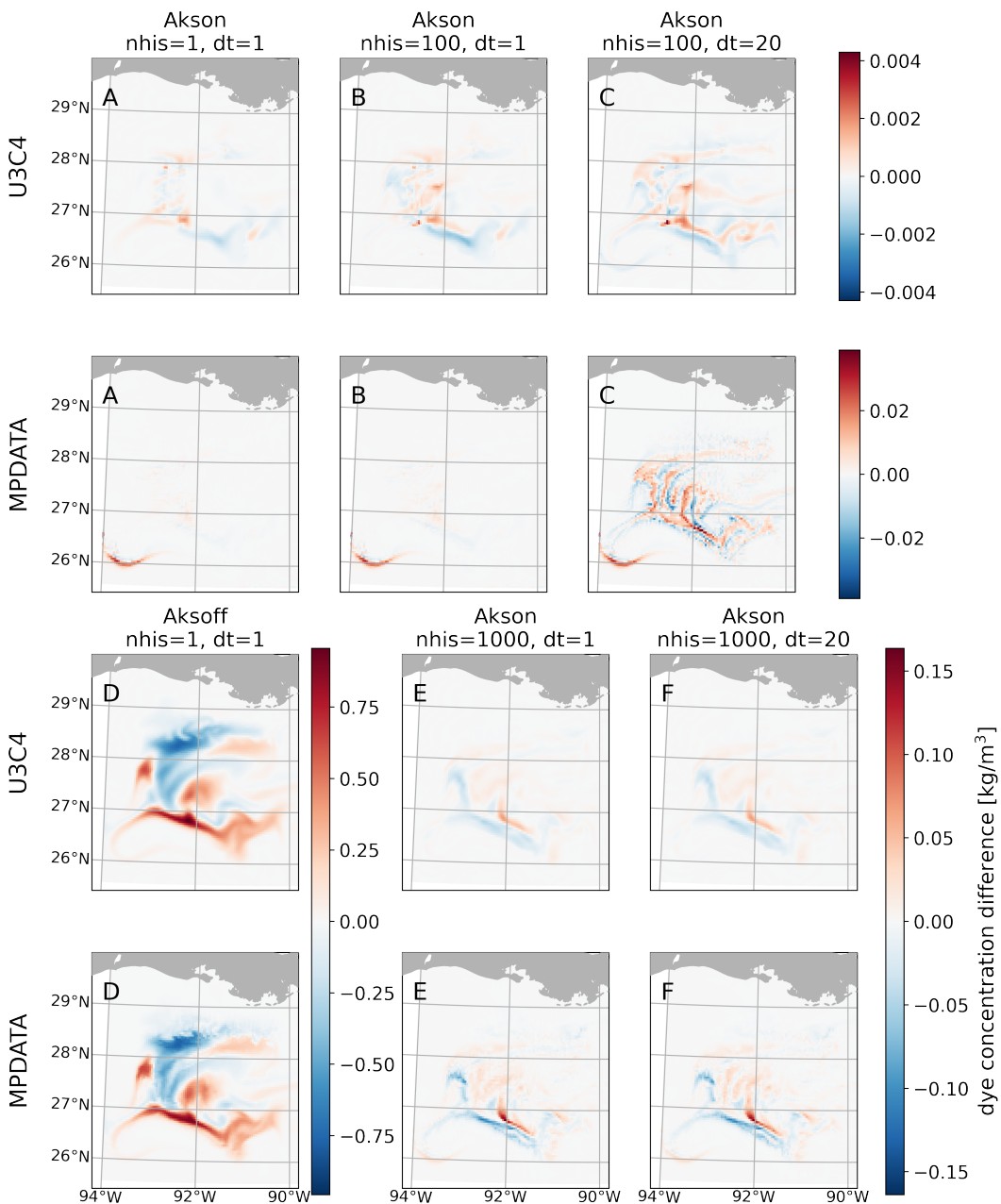

**Figure 3.** Instantaneous difference in dye concentration (online minus offline simulation) after about 13.2 days. Alternating rows show results from the two tracer advection schemes tested with the columns showing different experiments. All pairs of experiments except D forced the vertical salinity diffusion coefficient `Aks`. Experiments A vs. B show changing the forcing frequency of online output into the offline simulation, `nhis`, from 1 (every online time step) to every 100 online time steps the online time step. Experiment C shows additionally changing the offline time step to be 20 times the online time step. Experiments E and F show offline experiments forced with online output every 1000 online time steps with the offline timestep of the online (`dt=1`) or 20 times the online (`dt=20`) time step. Note that each colorbar has a different range of values.

Instantaneous differences in dye concentration demonstrate the spatial structure of the offline simulation errors (Figure 3). The structure changes not just with changes in the frequency of forcing in the offline simulation (`nhis`) and offline time step (`dt`), but also with the tracer advection scheme used. Comparing the top two rows in Figure 3 we see that the error in the MPDATA simulations tends to be more localized when compared with the U3C4 simulations. The magnitude of error increases with both decrease in forcing frequency and increase in time step for the offline simulations (moving from subplots A to C); in particular, the MPDATA simulation shows much more widespread spatial structure in the errors with `dt`=20 (subplots C). Subplots E and F show fairly similar structure across the simulations, though with larger errors for MPDATA. Subplots D show the much larger errors that result when the vertical salinity diffusion coefficient `Aks` is not forced in the offline simulation.

Skill scores (Equation 1) over time, demonstrating offline model accuracy, are shown in Figure 4, and a summary is shown in Table A1. Both tracer advection schemes (U3C4 and MPDATA) give highly accurate results (top), though U3C4 performs a bit better than MPDATA. When vertical salinity diffusivity, `Aks`, which controls the impact of sub-grid scale vertical mixing on the tracer field, is not forced (`Aksoff`), offline accuracy is reduced, though just two percentage points over 14 days compared to when it is forced. The impact of how often online model output is saved and input into the offline simulation, controlled by the `nhis` parameter, is almost negligible below 200 or 500 times the online time step (`nhis` 200, about 1.1 hours, and `nhis` 500, about 2.8 hours), but has increasing impact for less frequent online model output (higher values of `nhis`, middle subplot). This means that for the present model setup and region, frequency of online output higher than about 1–3 hours is not important. Results are relatively similar with `nhis` 1000 (about 5 hours), but accuracy decreases significantly as `nhis` increases beyond that. Context for the `nhis` values is given in Section 4.

The importance of `nhis` and the offline time step together for tracer advection scheme MPDATA is shown in Figure 4 (bottom). The largest control on the skill score is from `nhis` — the values shown demonstrate the spread from the highest accuracy to several levels down (`nhis` 200, 500, and 1000 times the online time step, or about 1, 3, and 5.5 hours, respectively). For each `nhis` value, three different offline time steps, `dt`, are shown (`dt` of 1, 10, 20 times the online time step). Accuracy decreases with increasing offline time step, but in different relative amounts that depend on the `nhis` value. For `nhis` of 200 and 500, there is more impact from the change in offline time step `dt` than from the `nhis` value. However, for `nhis` 1000, the `dt` values do not strongly impact the results. Offline time step results for U3C4 simulations are not shown because the time step does not strongly impact results for any `nhis` values.

Several issues are demonstrated in Figure 5. First is an example of model performance for skill score based on persistence (Equation 2). Model performance is similar, though a little lower, when assessed using the persistence skill score as compared with the regular skill score, so it is only shown here. This tells us that the offline model does indeed provide more benefit than simply persisting the initial condition. Next is a demonstration of offline accuracy compared to online output when different tracer advection schemes are used (middle). For reference, simulations forced with the same tracer advection schemes both on and offline are shown as well (U3C4, black solid, and MPDATA, black dashed). We find a significant decrease in offline model accuracy when the offline advection scheme does not match the online scheme, because different numerical schemes have different numerical dispersion and diffusion properties leading to differences in tracer advection. For comparison, the "skill score" comparing online U3C4 and online MPDATA output (gray dashed) is shown. The online-online comparison for the two

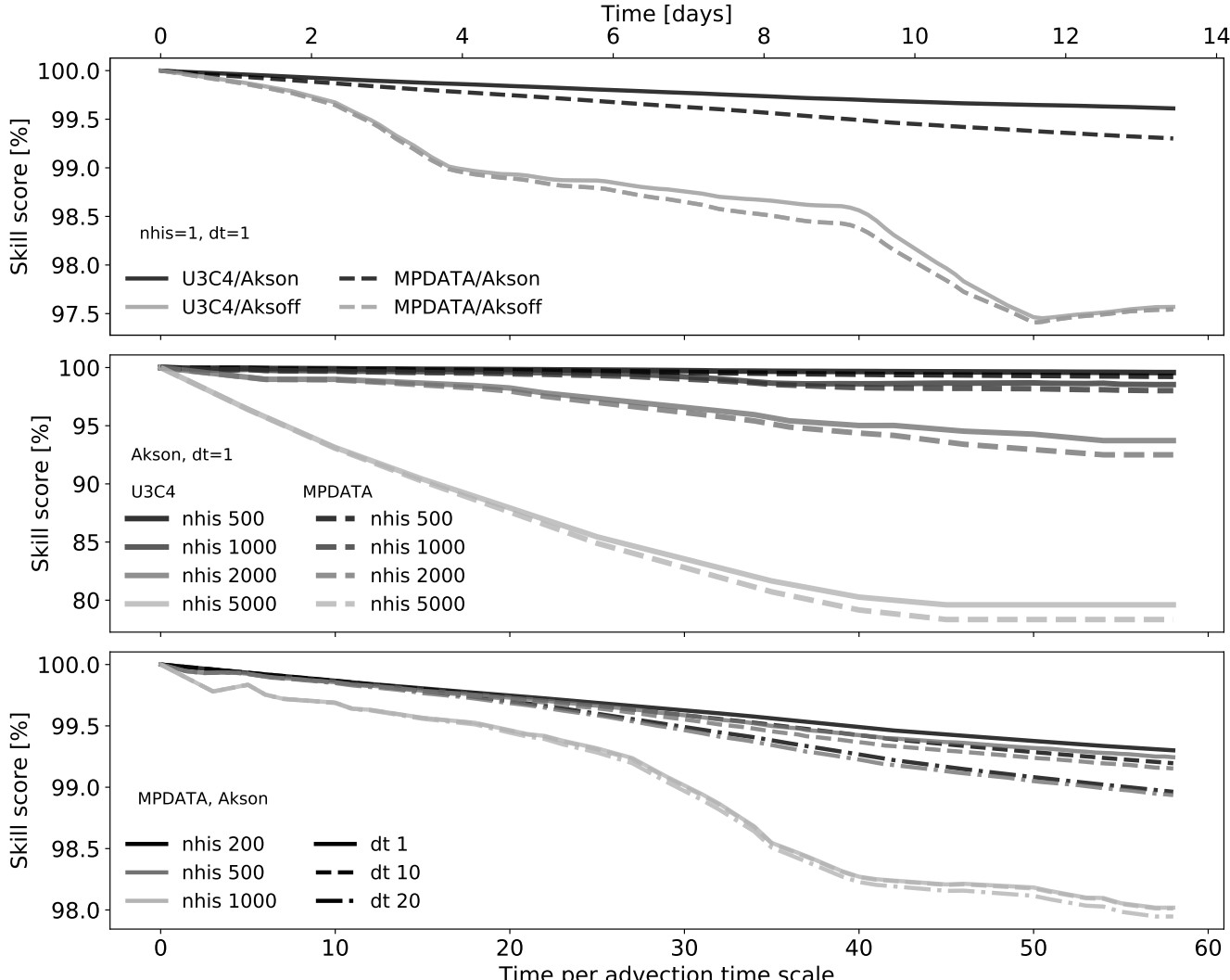

**Figure 4.** Skill scores for several subsets of offline simulations. (Top) Performance between tracer advection schemes `MPDATA` and `U3C4` and whether the vertical salinity diffusion coefficient `Aks` is forced with the online simulation ("on") or a constant background number ("off"). These cases also have `nhis=1` (online output was saved each time step) and `dt=1` (offline time step matched online time step). (Middle) Performance between `MPDATA` and `U3C4` advection schemes with `nhis` values varying. These cases also have `Aks` forced from online and `dt=1`. (Bottom) Performance for varying `nhis` and `dt` parameters, where `MPDATA` is used and `Aks` is forced from the online case. All offline simulations here forced by `his` files.

schemes has comparable performance, though lower; it is not clear there is a reason that the on and offline combinations should be better or worse than this, but the issue was not further explored. The best fidelity to an online simulation will be found by forcing offline with the same tracer advection scheme as online. Also, forcing offline with a different tracer advection scheme from online will give results that are different from the online results on the order of the difference between the results of the different tracer advection schemes themselves. Finally, the significant impact of using single precision output is demonstrated (bottom); it is best to save online model output for forcing offline simulations with double precision.

Several other issues were investigated but not plotted (they can be seen in the paper GitHub repository). Passive tracers are conserved in online ROMS simulations (Shchepetkin and McWilliams, 2005); offline simulations also conserve tracers. Only small differences were found between forcing offline simulations with snapshots (`his` files) or averages between time steps (`avg` files) from online simulations. Finally, for simulations in which a realistic `Aks` field was not forced, the background value used for `Aks` was varied; we found this did not impact results.

An overview of results is shown in Figure 6. The objective of this figure is to display the competing factors – computation time ($x$-axis) and storage required ($y$-axis) – that will ultimately determine offline accuracy (colored markers). Skill scores are shown for four subsets of simulations: tracer advection scheme `U3C4` with (diamonds) and without (downward-pointing triangles) `Aks` realistically forced, and tracer advection scheme `MPDATA` with (squares) and without (upward-pointing triangles) `Aks` realistically forced. The best compromise of storage, computational time, and skill score is where the skill score is still high – in one of the top classes, but with the lowest storage and time requirements. For the present set of simulations, this occurs for `U3C4` with realistic `Aks` for `nhis` of 200 (more conservative) or 500 and `dt` of 20, and for `MPDATA` with realistic `Aks` for `nhis` 200 and `dt` of 5. Simulations in which `Aks` is not forced always have lower accuracy and the small storage saving is probably not worth the loss, however, there may be circumstances in which online `Aks` is not available.

## 3.2 At-depth realistic Gaussian

The biggest difference in the second set of simulations (initialization shown in Figure 2) compared with the first is the variation in the vertical direction: a dye blob was initialized at a particular depth instead of throughout the water column. A skill score comparison between 13 and 14 days indicates that the good resolution experiment (`nhis=100`, or online output forced in the offline simulation every ∼30 min) had about the same skill score of 99.6% as the comparable previous numerical experiment skill score. However, the low resolution test case (`nhis=1000`, or online output forced every ∼5.5 hrs) had a much lower skill score of 70% compared with the first test case of 98.5%, possibly indicating a compensatory effect in the first set of experiments in the vertical direction. That is, dye may have been transported vertically inaccurately in the first set of lower resolution experiments but since the whole water column had dye in it, it may have still given better skill scores than if the dye patch was instead discrete.

Spatial differences in the accuracy of the experiments are shown for depth slices (Figure 7) and cross-sections (Figure 8). The dye in the good resolution cases stays close to the online simulation, with small differences in the percent error near where the dye encounters the bathymetry on the west end of the blob (Figure 7D and F, noting that the values in D have been multiplied by 100 to be visible). The low resolution case is qualitatively similar to the online case, but the difference (Figure 7E) shows

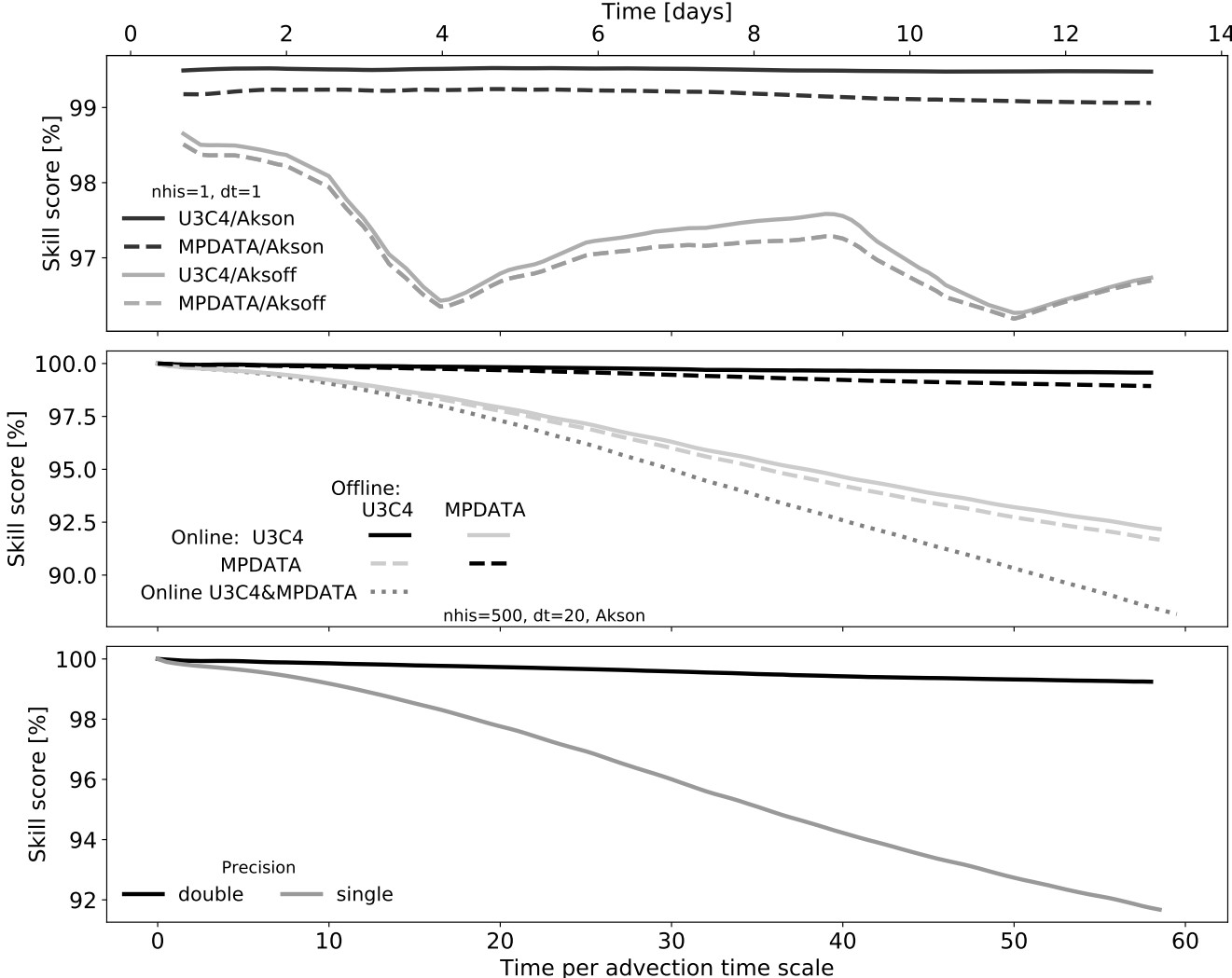

**Figure 5.** (Top) Skill score compared with persistence is shown for combinations of tracer advection scheme and whether online `Aks` is forced. The simulations shown are the same as in Figure 4 (top). (Middle) Comparison of skill score simulations forced by several combinations of tracer advection schemes. Combinations are: online simulation using `U3C4` with offline simulation using `U3C4` (black solid line), online simulation using `U3C4` with offline simulation using `MPDATA` (gray solid), online `MPDATA` with offline `U3C4` (gray dashed) and with offline `MPDATA` (black dashed), and comparison between results from online `U3C4` and online `MPDATA` (gray dotted). (Bottom) Skill score for double compared with single precision online output.

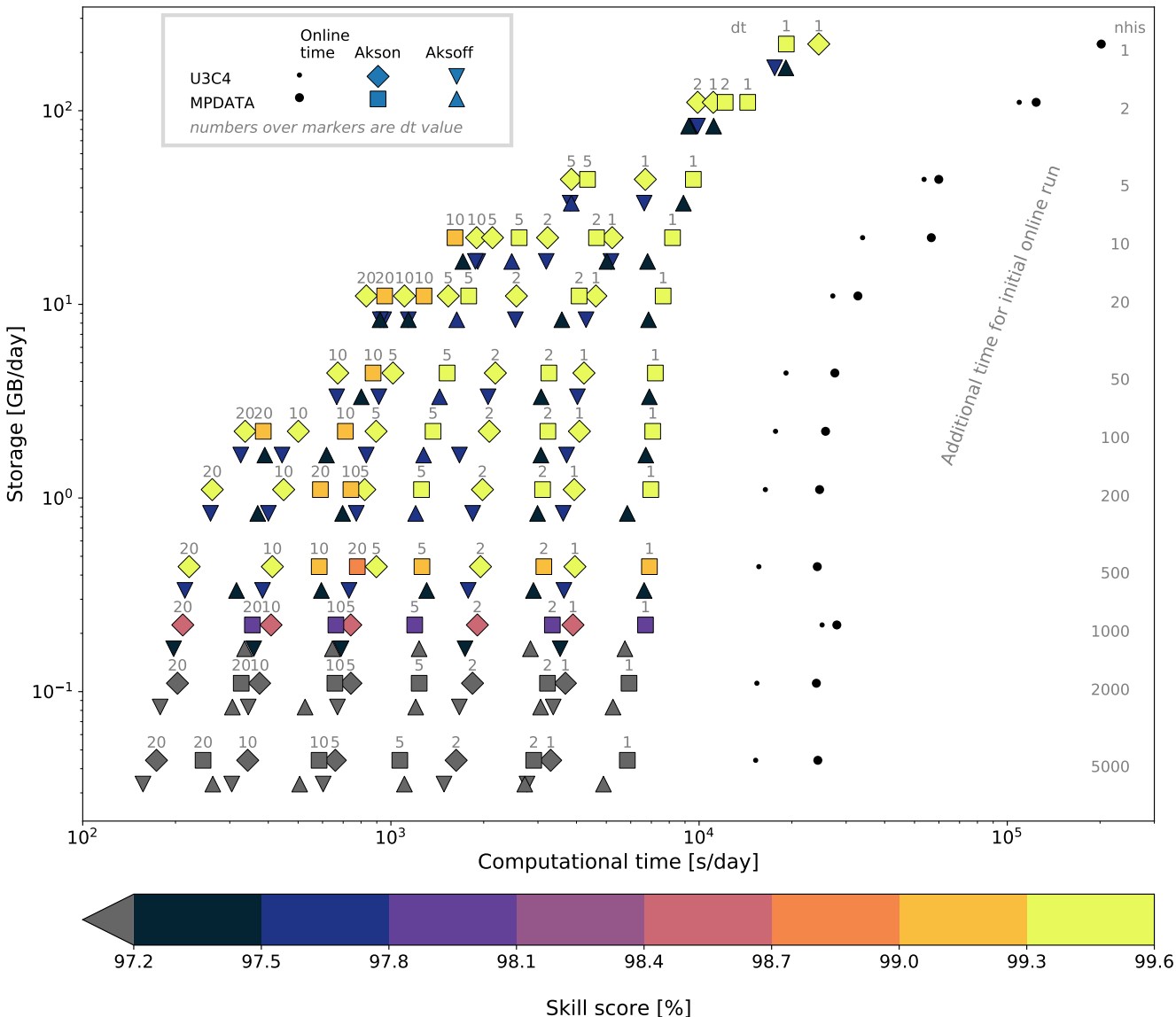

**Figure 6.** Summary of skill score results. Shown are the offline computational time per simulation day ($x$-axis), storage required for the online simulation per simulation day ($y$-axis), and the skill score after about 13.5 days of simulation when forcing with snapshots (`his` files) for a range of `nhis` and `dt` values (colored markers, with one set for forcing `Aks` or not, and which tracer advection scheme is used). `nhis` values for rows are indicated on the right hand side of the plot and `dt` values are above each pair of markers. Values below 97.2% are colored gray. Computational time required for the online simulation is shown separately with black markers.

patches of large disagreement. The disagreement is further demonstrated in the low resolution case percent error (Figure 7G) with a swath of 1–10% error across the full dye feature.

Results are similar for the vertical cross section (Figure 8). The differences in the offline and online dye field are very small in the good resolution case – it has been multiplied by 500 to appear on the same colorbar as the low resolution case. In the low resolution case, the offline dye has been transported both up and down more than in the online case.

## 4   Discussion

The context of the performance difference found as a function of `nhis` values (Figure 4 (middle, bottom)) can be considered as the impact of loss of energy represented in the system (an approach also used by Qu and Hetland (2019)). For example, Figure 9 shows the power spectral density of the online simulation speed from near the middle of the dye patch. The output frequency, `nhis`, from the online simulation controls how much energy of this spectrum the offline simulations receive, and therefore how much of the system's energy is represented offline. The amount of energy missing can be seen visually by the overlaid lines representing different output frequencies, `nhis`. Skill score results (Figure 4) show that accuracy decreases as `nhis` values increase starting at `nhis` of 200 or 500 (about 1 to 3 hours), which correspond to between 1 and 5% of the total energy being lost to subsampling the output.

Comparing a relevant dynamical timescale to `nhis` is another way to provide context for its impact on offline accuracy. A previous study evaluating an offline tracer from MITgcm model output found that for their global-scale model, the inertial period controlled the output rate necessary for robust results (Hill et al., 2004). We find an analogous result here, though the relevant timescale is the advection timescale (see Section 2.1). The advection timescale for this regional model is about 20,000s, which corresponds to an output rate from the online model of `nhis`=1000 times the online time step, which is indeed the turning point for clear degradation in offline model accuracy we find (Figure 6).

We should expect that the offline time step is controlled by the horizontal Courant number and that our results destabilize as the number increases toward 1. An estimate of the horizontal Courant number, with largest horizontal velocity of 1 and smallest horizontal cell width as about 3800 m, for the offline time steps gives a range from 0.005 for offline time step matching the online time step up to about 0.1 and 0.25 for offline time step `dt` of 20 and 50, respectively. Simulations gave reasonable results for `dt` of 20, but not 50.

## 5   Conclusions

This paper presents a description and evaluation of an offline tracer advection model developed within ROMS. The advantage of this is the ease and consistency with which ROMS users can use existing model output to force offline tracer simulations at low computational cost. The main approach of the offline model is to force variables `zeta`, `u/v`, and `ubar/vbar` from an online simulation as climatology; normally climatology would be used in a ROMS simulation to nudge boundary conditions toward mean values, but in this case all grid cells are fully forced. Additionally forcing the vertical salinity diffusivity, `Aks`,

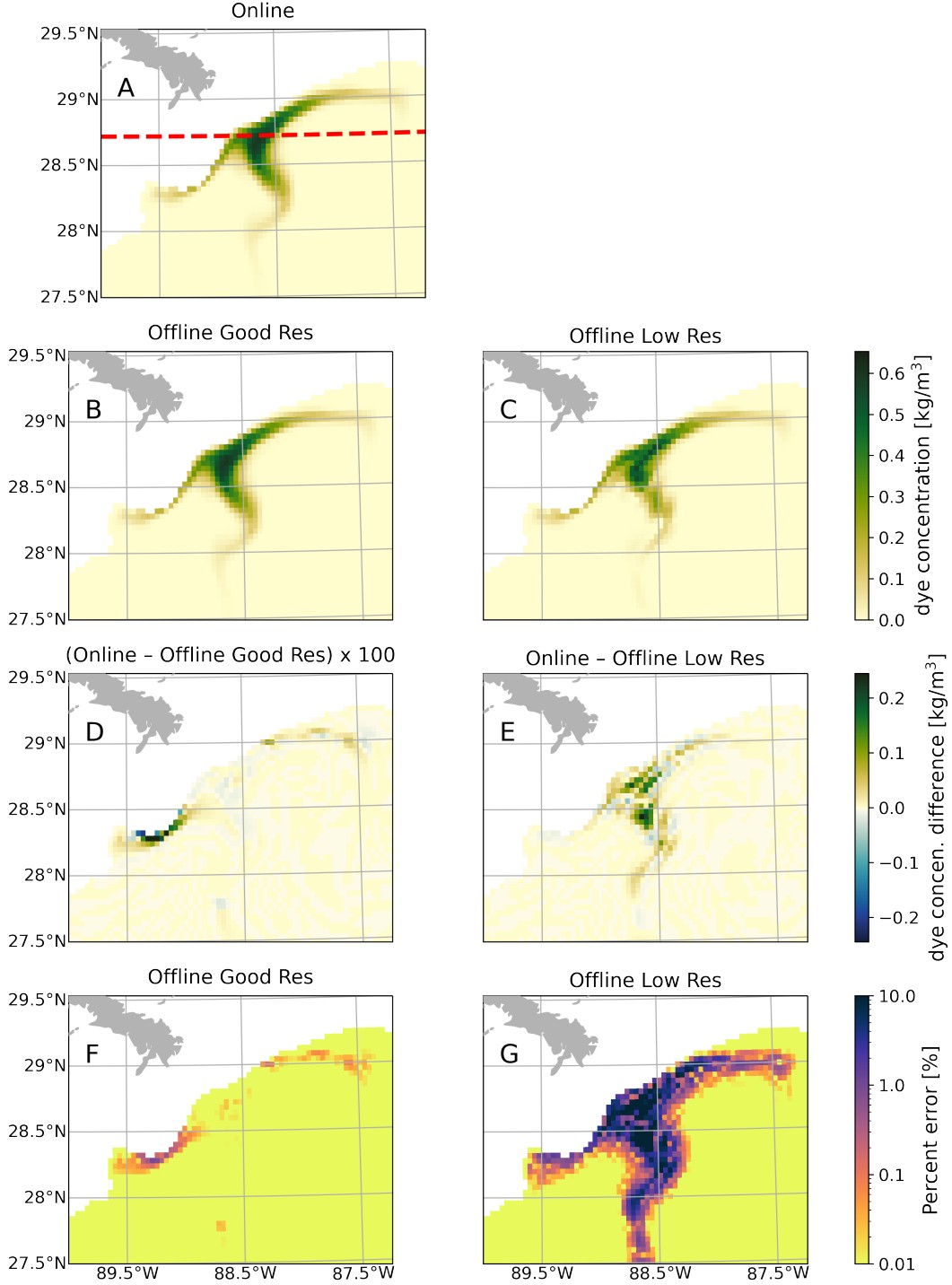

**Figure 7.** Snapshots of the dye at 13.75 days for the online (A), good resolution offline (B) and low resolution offline (C) simulations. The difference in dye concentration for the online and offline cases at the same time for the good (D) and low (E) resolution experiments. Percent error for the good (F) and low (G) resolution offline cases is also shown. Subplot (A) also indicates the slice location shown in Figure 8. Note that values in (D) have been multiplied by 100 to be visible on the same colorbar as (E) since the differences are so small.

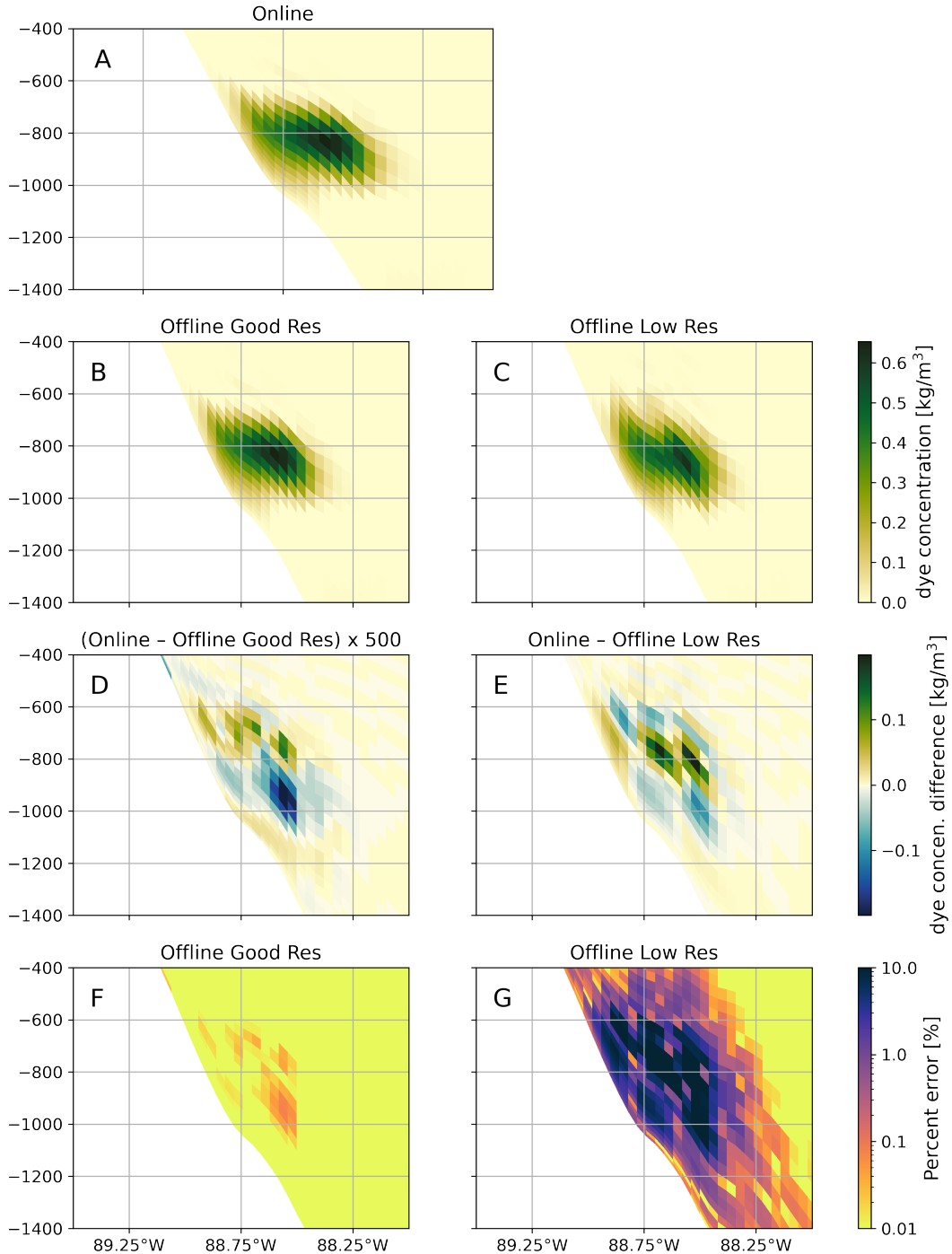

**Figure 8.** Vertical cross section comparisons of the online and offline simulations; the cross section location is indicated in Figure 7(A). Snapshots at 13.75 days are shown for the online (A) and offline good (B) and low (C) resolution cases. Differences at the same time are shown in (D) and (E). Percent error is shown in (F) and (G). Note that values in (D) have been multiplied by 500 to be visible on the same colorbar as (E) since the differences are so small.

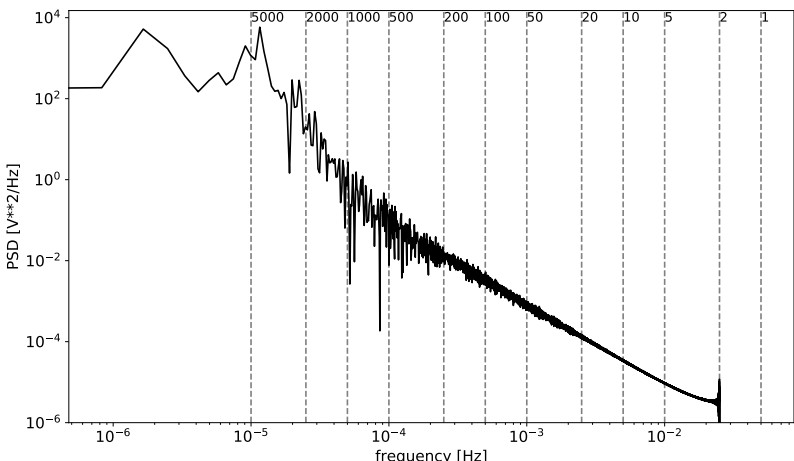

**Figure 9.** Power spectral density for speed at a single location near the center of the dye patch. Overlaid (gray dashed) are lines marking frequencies at which online model output was saved for forcing offline simulations; these are marked with their corresponding `nhis` value.

improves model accuracy. It is also important that the online simulation output used to force the offline simulation have double precision.

We tested two tracer advection schemes, `MPDATA` and `TS_U3HADVECTION` with `TS_C4VADVECTION` (3rd-order upstream horizontal advection and 4th-order centered vertical advection, called `U3C4` here) in a regional simulation of the north Gulf of Mexico, and found that the offline simulations are able to reproduce online simulations to high accuracy. The most important control differentiating offline accuracy was the `nhis` parameter describing how often online simulation output was saved, as a multiple of the online time step, to be input into the offline simulation. For both tracer advection schemes and with

`Aks` forced, the offline simulations showed high accuracy up to `nhis=200` or 500, about 1.1 and 2.8 hours. This is consistent with requiring temporal information at a rate higher than the relevant dynamic timescale, in this case an advection timescale approximated here as approximately equivalent to an `nhis` value of 1000. The offline time step `dt` was not an important choice for offline simulations run with `U3C4`, as long as it was under about 50 (all had skill scores of 99.6% after 14 days). However, for `MPDATA` offline simulations were highly accurate with a time step 5 times the online time step up to `nhis=200`,

with some dependence on the offline time step.

     A second set of simulations were run to demonstrate performance in a more realistic, application-driven experiment, in this case with a discrete blob of dye at depth. The good resolution case with online forcing at a frequency of `nhis=100` (about 30 min) was very accurate, with a similar skill score to the original comparable offline U3C4 experiment run of 99.6%. The low resolution experiment of `nhis=1000` (about 5.5 hr) gave worse results than the comparable previous simulation, implying

that the vertical direction indeed is important and can behave distinctly from the horizontal. Overall, the results show that it is possible to get high fidelity results in offline tracer simulation with this code.

*Code and data availability.* The current versions of the related code and data are available online, all under the MIT license: the offline tracer model https://github.com/kthyng/COAWST-ROMS-OIL, the analysis for this manuscript https://github.com/kthyng/offline_analysis, run files for online simulations https://github.com/kthyng/oil_03, run files for offline simulations https://github.com/kthyng/oil_off. The exact version of the model used to produce the results used in this paper is archived on Zenodo (doi: 10.5281/zenodo.3991810), as are scripts to run analysis and produce the plots for all the simulations presented in this paper (doi: 10.5281/zenodo.4278115), run files for online simulations (doi: 10.5281/zenodo.3991823), and run files for offline simulations (doi: 10.5281/zenodo.3991826). Input data to run the model are available both on figshare (doi: 10.6084/m9.figshare.c.5097350.v1) and through GRIIDC (doi: 10.7266/YF0QPBFC). Simulation output from the online and offline simulations is available through GRIIDC (doi: 10.7266/7R0N3FX4).

# Appendix A: Table of skill scores

**Table A1.** Final skill score (percent) of offline simulations after 14 days, sorted by `nhis` and `dt` values, tracer advection scheme, and if `Aks` is forced.

| advect | | MPDATA | | U3C4 | |
|---|---|---|---|---|---|
| Aks | | off | on | off | on |
| nhis | dt | | | | |
| 1 | 1 | 97.5 | 99.3 | 97.6 | 99.6 |
| 2 | 1 | 97.5 | 99.3 | 97.6 | 99.6 |
| | 2 | 97.5 | 99.3 | 97.6 | 99.6 |
| 5 | 1 | 97.5 | 99.3 | 97.6 | 99.6 |
| | 5 | 97.6 | 99.3 | 97.6 | 99.6 |
| 10 | 1 | 97.5 | 99.3 | 97.6 | 99.6 |
| | 2 | 97.5 | 99.3 | 97.6 | 99.6 |
| | 5 | 97.6 | 99.3 | 97.6 | 99.6 |
| | 10 | 97.5 | 99.2 | 97.6 | 99.6 |
| 20 | 1 | 97.5 | 99.3 | 97.6 | 99.6 |
| | 2 | 97.5 | 99.3 | 97.6 | 99.6 |
| | 5 | 97.6 | 99.3 | 97.6 | 99.6 |
| | 10 | 97.5 | 99.2 | 97.6 | 99.6 |
| | 20 | 97.5 | 99.0 | 97.6 | 99.6 |
| 50 | 1 | 97.5 | 99.3 | 97.6 | 99.6 |
| | 2 | 97.5 | 99.3 | 97.6 | 99.6 |
| | 5 | 97.6 | 99.3 | 97.6 | 99.6 |
| | 10 | 97.5 | 99.2 | 97.6 | 99.6 |
| 100 | 1 | 97.5 | 99.3 | 97.6 | 99.6 |
| | 2 | 97.5 | 99.3 | 97.6 | 99.6 |
| | 5 | 97.6 | 99.3 | 97.6 | 99.6 |
| | 10 | 97.5 | 99.2 | 97.6 | 99.6 |
| | 20 | 97.5 | 99.0 | 97.6 | 99.6 |
| 200 | 1 | 97.5 | 99.3 | 97.6 | 99.6 |
| | 2 | 97.5 | 99.3 | 97.6 | 99.6 |
| | 5 | 97.6 | 99.3 | 97.6 | 99.6 |
| | 10 | 97.5 | 99.2 | 97.6 | 99.6 |
| | 20 | 97.5 | 99.0 | 97.6 | 99.6 |

| advect | | MPDATA | | U3C4 | |
|---|---|---|---|---|---|
| Aks | | off | on | off | on |
| nhis | dt | | | | |
| 500 | 1 | 97.5 | 99.2 | 97.6 | 99.6 |
| | 2 | 97.5 | 99.2 | 97.6 | 99.6 |
| | 5 | 97.5 | 99.2 | 97.6 | 99.6 |
| | 10 | 97.5 | 99.2 | 97.6 | 99.6 |
| | 20 | 97.5 | 98.9 | 97.6 | 99.6 |
| 1000 | 1 | 96.9 | 98.0 | 97.2 | 98.5 |
| | 2 | 96.9 | 98.0 | 97.2 | 98.5 |
| | 5 | 96.9 | 98.0 | 97.2 | 98.5 |
| | 10 | 96.9 | 98.0 | 97.2 | 98.5 |
| | 20 | 96.9 | 97.9 | 97.2 | 98.5 |
| 2000 | 1 | 92.2 | 92.5 | 93.4 | 93.7 |
| | 2 | 92.2 | 92.5 | 93.4 | 93.7 |
| | 5 | 92.2 | 92.5 | 93.4 | 93.7 |
| | 10 | 92.2 | 92.5 | 93.4 | 93.7 |
| | 20 | 92.3 | 92.6 | 93.4 | 93.7 |
| 5000 | 1 | 78.2 | 78.3 | 79.4 | 79.6 |
| | 2 | 78.2 | 78.3 | 79.4 | 79.6 |
| | 5 | 78.2 | 78.4 | 79.4 | 79.6 |
| | 10 | 78.2 | 78.4 | 79.4 | 79.6 |
| | 20 | 78.3 | 78.4 | 79.4 | 79.6 |

**Appendix B: Explanation of code changes**

While preprocessor flags for offline simulations already existed in the ROMS/COAWST codebase, we found that the offline simulations did not work as desired. In this section, we describe changes made to the code base so that offline passive tracer advection works properly by receiving the necessary forced variables. Generally, the offline code works by forcing previously-simulated online model output input as climatological forcing. Typically, climatology would be used in a ROMS simulation to nudge boundary conditions toward mean values, but in this case all grid cells are fully forced.

Code changes were made to avoid repeating processes offline that were already included online. Initialization is now minimal for offline simulations (`ini_fields.F`), and initial values are replaced by the first time step read in from climatology. Updates to sea surface height `zeta` (calls to `ini_zeta` and `set_zeta` in `main3d_offline.F`) have been removed since the variable is directly forced in the offline simulation. Boundaries are not forced in the offline case (except for the passive dye field): horizontal indices now start 1 earlier and end 1 later in each tile so that climatology is read into ghost cells in place of boundary conditions (`set_data.F`). The remaining processes are controlled through the user input file and pre-processor flags (Section C).

The offline simulation is missing much of the complex time stepping in an online ROMS simulation due to the missing numerics, leading to necessary code adjustments (`set_data.F`). Climatology for 3D variables (`u`/`v`, `salt`/`temp`, `tke`/`gls`) are read into earlier time indices (`nrhs` instead of `nnew`) to account for this, eliminating a time shift that otherwise occurs. Model output for the subsequent time step are read in from climatology and saved in available time indices for several variables (`zeta`, `Aks`, `Akt`) to be used later in the time loop. In the online simulation, `zeta` is normally updated mid-time loop with the fast time stepping value. To approximate this behavior, the two time steps of `zeta` are averaged into variable `Zt_avg1` (new function `set_avg_zeta`). Calculations of vertical layer thickness `Hz` and mass fluxes `Huon`/`Hvom` for the subsequent time step are made mid-time loop in `set_depth.F`. New functions `set_massflux_avg` and `set_massflux_avg_tile` were added to `set_massflux.F` to average `Huon`/`Hvom` values to be used subsequently in `step3d_t.F` where the tracer is advected. This change matches the online case to floating point round-off error.

The `OFFLINE` preprocessor flag with `OFFLINE_TPASSIVE` compiles the necessary code to run offline tracer advection (more details in Section C). These flags already existed, but changes to the code for the present project were made under these flags. Other available offline preprocessor flags include:

- `omega` (`OCLIMATOLOGY`): already existed, does not impact offline tracer advection. Reads in climatology for S-coordinate vertical momentum component.

- `Aks` (`AKSCLIMATOLOGY`), `Akt` (`AKTCLIMATOLOGY`), `Akv` (`AKVCLIMATOLOGY`), or all three `Aks`, `Akt`, `Akv` (`AKXCLIMATOLOGY`): new flags. `Aks`, the vertical salinity diffusion, impacts accuracy of the offline tracer (Section 3). While `Akt`, the vertical temperature diffusion, does not impact offline tracer advection, it is used for offline floats (`OFFLINE_FLOATS`) if vertical walk (`FLOATS_VWALK`) is activated. `Akv`, the vertical viscosity, does not impact offline passive tracer advection.

– TKE (`TKECLIMATOLOGY`, turbulent kinetic energy), GLS (`GLSCLIMATOLOGY`, generic length scale), or both (`MIXCLIMATOLOGY`): new flags. These do not impact offline passive tracer advection.

     – `salt` and `temp` (`ATCLIMATOLOGY`): new flag. Also impacted by `LtracerCLM` in the input file. While these do not impact offline tracer advection, they may be used for other modules such as oil modeling with offline floats.

To fix a problem with reading in the climatology at the correct time step, a condition was added (`get_2dfld.F` and
`get_3dfld.F`) that compares the differences in times to being less than half a time step, avoiding any problems with numerical precision.

`omega`, the mass flux perpendicular to the local s-coordinate, was already setup to be read in through climatology with the `OCLIMATOLOGY` flag, but results did not match on and offline. The lower vertical index in the call for omega in `get_data.F` was 1, which is used for `rho` grids instead of the w-grid `omega` is actually on, which starts at index 0.

**Appendix C: How to set up simulations**

Requirements and considerations for setting up online and offline simulations in ROMS or COAWST with the offline passive tracer advection code are provided below.

**C1    Online**

**C1.1    Input file**

In the project input file (the `*.in` file, for example, https://github.com/kthyng/oil_03/blob/master/External/ocean_oil_03.in), the items below should be considered in addition to the typical input parameter selections:

     – Choose whether to save output as snapshots at a single time or averages across time intervals (ROMS `his` vs `avg` files). Your choice will be used to force the offline simulation. Present results show this choice does not significantly change results. We recommend using `his` files in the absence of any other preference since otherwise it is necessary to include
the initial file prepended to the input `avg` file in `CLMNAME`.

     – Output necessary variables for forcing the offline simulation. Variables `zeta`, `ubar`, `vbar`, `u`, `v` are required for forcing the offline simulation, and `Aks` is optional for improved accuracy in the offline simulation (though it increases amount of storage required).

     – Choose output frequency (parameter `NHIS` for `his` files or `NAVG` for `avg` files). This is how often ROMS will save
output to a `his` or `avg` file, as a multiple of the time step, and in turn this is what will be used to force the offline simulation. Important considerations for this selection include acceptable simulation runtime and storage requirements. Figure 6 gives a paradigm from which to decide this for simulations in general. In the present study, for `U3C4` there was a drop in performance below an output frequency of 500 times the online time step, and below 200 or 500 times for `MPDATA`. These choices will vary for a given model setup and accuracy needs.

– For this online simulation, point to file `varinfo-online.dat` for `VARNAME`, which is a typical, unchanged file. This has been provided in the code repository: https://github.com/kthyng/COAWST-ROMS-OIL/blob/master/ROMS/ External/varinfo-online.dat.

## C1.2    Header file

In the project header file (the `*.h` file, for example, https://github.com/kthyng/oil_03/blob/master/Include/oil_03.h), the addi-
tional flags below should be considered:

     – Choose a tracer advection scheme. We tested two schemes and found both accurately reproduced the online results offline, though `U3C4` performed slightly better. Note, however, that online tracer advection performance itself depends on the dynamics involved; more information is available in Kalra et al. (2019). Also note that `MPDATA` requires more runtime than `U3C4` (Figure 6).

– Use `OUT_DOUBLE` to output results with double precision to significantly improve your accuracy, though increase storage required (Figure 5).

## C2    Offline

## C2.1    Input file

In the project input file (the `*.in` file, for example, https://github.com/kthyng/oil_off/blob/master/External/ocean_oil_offline.
in), the items below should be considered in addition to the typical input parameter selections:

     – The output frequency (`NHIS` or `NAVG`) will not impact your offline simulation performance, but should be chosen to well-represent the dynamics in your model.

     – A reasonable choice for the offline simulation time step `DT` is a multiple of the online time step. Some testing for your model setup is warranted. The present study found that time step was not important for the U3C4 tracer scheme
combination — a `DT` of 20 times the online time step gave as good of accuracy as the online time step itself. However, for `MPDATA`, only using the online time step gave the highest accuracy; for the next level down of accuracy a time step of 10 times the online time step was adequate (Figure 6). Note also that the offline time step needs to factor evenly into the online output frequency, and the offline time step cannot be larger than the online output frequency.

     – All physics should be off in the offline case, except for anything directly impacting the offline tracer field (`dye_01`)
itself, because it is included in the online output. This implies:

         – Boundaries should all be closed except for offline tracer fields (*e.g.*, parameter `LBC(isFsur)`).

         – Turn off river forcing and other sources or sinks that were forced in the online simulation.

         – Do not force winds, bulk fluxes, etc, from the online simulation.

- Do not nudge to climatology, even if used in the online simulation (climatology, the output from the online simulation, will be entirely enforced).

- Turn on flags for climatology forcing for sea surface height (`LsshCLM`), and 2D (`Lm2CLM`) and 3D momentum (`Lm3CLM`). Turn on salt and temperature flags (`LtracerCLM`) if you want to read them in (see header section next).

- Only need to output sea surface height (zeta) and the offline dye(s) (`dye_01`) – other fields are best used directly from the online simulation (the vertical velocity $w$ for example is not calculated properly in the offline simulation). The sea surface height is necessary to properly calculate tracer advection fluxes.

- Input as the climatology forcing (`CLMNAME`) the online model output. If forcing with an `avg` file from the online simulation, it is necessary to place a file containing the initial conditions first; this is possible by inputting a list of file names.

- For this offline simulation, point to file `varinfo-offline.dat` for `VARNAME`, which has been edited to include the new variables that can be input as climatology and so that all climatology time variables are named `ocean_time`. The latter change allows for the online output to be input directly offline as climatology without processing the file to rename variable attributes. The file has been provided in the code repository: https://github.com/kthyng/COAWST-ROMS-OIL/blob/master/ROMS/External/varinfo-offline.dat.

## C2.2 Header file

In the project header file (the `*.h` file, for example, https://github.com/kthyng/oil_off/blob/master/Include/oil_offline.h), the additional flags below should be considered:

- Use the `OFFLINE` flag for any offline simulation, and additionally the `OFFLINE_TPASSIVE` flag for offline tracer advection.

- For best results, use the same tracer advection scheme as the online run. The schemes do not have to match but the skill score between the simulations will diminish substantially (Figure 5) since they do not use the same numerics. We did not test other tracer advection schemes but we have no reason to think they will not work offline.

- Forcing the vertical salinity diffusivity `Aks` as predicted by the online simulation gives better offline accuracy than not, though requires storing it from the online case. This can be forced with the `AKSCLIMATOLOGY` flag. More information on the offline flags is available in Section B.

*Author contributions.* KMT edited the code, performed final simulations and analysis, and wrote the text. DK, VRX, and LQ edited the ROMS code and ran simulations. XC created the regional model setup in ROMS. RDH participated in discussions with ideas.

*Competing interests.* The authors declare that they have no conflict of interest.

*Acknowledgements.* This research was made possible by a grant from The Gulf of Mexico Research Initiative. In addition to the locations
noted under "Code and data availability," data are publicly available through the Gulf of Mexico Research Initiative Information & Data
Cooperative (GRIIDC) at https://data.gulfresearchinitiative.org (doi: 10.7266/YF0QPBFC, doi: 10.7266/7R0N3FX4).

Thank you to the Texas A&M High Performance Research Computing center for hosting simulations.

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
