# Peer review of "Performance of offline passive tracer advection in ROMS (v3.6, revision 904)"

_Geoscientific Model Development, 2020_

## Referee Comment (RC1) · Anonymous Referee #1 · 12 Oct 2020

This paper mentions a methodology that was applied to a commonly used ocean model ROMS for running it offline in order to save computational time. The results of the study are somewhat intuitive i.e. a frequency of output that corresponds with a time step that can resolve advection time scales, forcing realistically and using double precision would help in the most accurate solution. It is unclear that it would add significant scientific value to the existing literature although it could be a good case study for folks trying to model similarly. It would then require that the authors discuss another example, perhaps something more application oriented besides the test case mentioned in the paper. That would also prove the repeatability of some of the key conclusions. It would be also good to add the equations that are solved via a schematic or a written description when the model is simulated in an offline manner. That would help modelers

using other type of models get ideas from the paper to broaden its appeal to a wider audience.

Some minor corrections 1. page 1Line 15-> change time savings to improved computational efficiency 2. page 2 line 22-> "showed good accuracy by hill et al." Is there a specific result that Hill showed that can be summarized here. 3. Besides the 84 proc to 28 proc change, not sure the rest of the paragraph is needed. 4. Figure 4 is hard to interpret. what is the significance of y axis representing storage . the x axis should be computational time.

good job with the appendices

———————————————————

---

## Referee Comment (RC2) · Anonymous Referee #2 · 19 Oct 2020

General Comments:

This paper provides an excellent description of a new method for running ROMS, a open-source, commonly used hydrodynamic ocean model, offline. The paper does not, to the best of my understanding, represent a huge advance the field of numerical modeling itself, but it does provide documentation of a new tool available to the scientific community. This is consistent with the goals of the GMD journal.

Overall, I consider the paper to be excellent in scientific quality and presentation quality, and moderate on scientific significance. The archiving of all relevant files to reproduce the results implies it has excellent reproducibility.

The paper could be improved by considering non-spatially averaged skill scores. In

most coastal systems, the dynamics, as well as the representative length scales and temporal scales, vary in space (and time). The extent to which this affects the skill scores in different areas of the grid would be of much interest to readers and possible users of this software.

It would also be useful to demonstrate that this method works for more than one model configuration. Different pre-processing choices, grid configurations, open boundary conditions, etc. may all impact the ability for this software to be implemented by other ROMS users.

Specific Comments & Technical Corrections:

1. Lines 63-5: This sentence is confusing as written. I suggest deleting "as opposed to" and breaking the sentence into two sentences.

2. Can you include a map showing the difference in tracer concentration among different model runs so that users can visually see the magnitude and spatial variability of the error of the offline simulations, compared to the online simulations?

3. Figure 4 contains a lot of important information, but was difficult to understand. I suggest considering removing the 'dt' from the figure. If needed, this could be included in a subplot. Also, including nhis as a 2nd y-axis instead of numbers on the plot, would be useful for orienting the reader. Finally, drawing a box around the legend would help readers more readily separate it from the rest of the text in the figure. If the dt's are kept in the figure, please include them in the legend.

---

## Author Comment (AC1) · 17 Nov 2020

Thank you for your review. We appreciate the time you spent to do this. Our responses are below. The reviewer comments are included in italics with our responses after each.

*This paper mentions a methodology that was applied to a commonly used ocean model ROMS for running it offline in order to save computational time. The results of the study are somewhat intuitive i.e. a frequency of output that corresponds with a time step that can resolve advection time scales, forcing realistically and using double precision would help in the most accurate solution. It is unclear that it would add significant scientific value to the existing literature although it could be a good case study for folks trying*

[Figure]

*to model similarly. It would then require that the authors discuss another example, perhaps something more application oriented besides the test case mentioned in the paper.*

Another experiment has been added to the manuscript that is more application oriented, as suggested.

*That would also prove the repeatability of some of the key conclusions. It would be also good to add the equations that are solved via a schematic or a written description when the model is simulated in an offline manner. That would help modelers using other type of models get ideas from the paper to broaden its appeal to a wider audience.*

No equations were modified for this offline tracer advection scheme. The changes made to ROMS were all to be able to force additional variables as climatology. The tracer advection is forced exactly as normal in ROMS, but the velocity fields advected the tracer are input as climatology (previously saved from an online run) instead of calculated at the time by ROMS.

*Some minor corrections*

1. *page 1 Line 15-> change time savings to improved computational efficiency* Done.

2. *page 2 line 22-> "showed good accuracy by hill et al." Is there a specific result that Hill showed that can be summarized here.* This sentence has been added: "For a set output frequency, an offline time step of8 times the online time step gave a skill score of over 98

3. *Besides the 84 proc to 28 proc change, not sure the rest of the paragraph is needed.* The goal with this is to be open about analysis tools used and also to give appropriate credit.

4. *Figure 4 is hard to interpret. what is the significance of y axis representing stor-*

*age . the x axis should be computational time. good job with the appendices*
The x-axis is computational time per simulation day. The label and caption have
been modified to say this more explicitly. The y axis is storage required to run on
the online simulation which is then forced in the offline simulation (per simulation
day), and it is one of the tradeoffs required when deciding how to run an offline
simulation, along with computational time. This figure has been modified to try to
be easier to see what is important but altering text lightness, etc.
221, 2020.

---

## Author Comment (AC2) · 17 Nov 2020

Thank you for your review! We appreciate your time to do this. Our comments are below. The reviewer responses are included in italics and our responses follow each.

*General Comments: This paper provides an excellent description of a new method for running ROMS, a open-source, commonly used hydrodynamic ocean model, offline. The paper does not, to the best of my understanding, represent a huge advance the field of numerical modeling itself, but it does provide documentation of a new tool available to the scientific community. This is consistent with the goals of the GMD journal. Overall, I consider the paper to be excellent in scientific quality and presentation quality, and moderate on scientific significance. The archiving of all relevant files to reproduce*

[Figure]

*the results implies it has excellent reproducibility. The paper could be improved by considering non-spatially averaged skill scores. In most coastal systems, the dynamics, as well as the representative length scales and temporal scales, vary in space (and time). The extent to which this affects the skill scores in different areas of the grid would be of much interest to readers and possible users of this software.*

A qualitative feel for the spatial structure of the offline simulation errors has been added through both the next response and though comment 2 below.

*It would also be useful to demonstrate that this method works for more than one model configuration. Different pre-processing choices, grid configurations, open boundary conditions, etc. may all impact the ability for this software to be implemented by other ROMS users.*

We present a new simulation using the same numerical model to try to address this point. The simulation is meant to emulate a "real world case" by being at depth and more localized. Skill scores (averaged over space) and percent errors (shown for a snapshot in time and in both planview and a vertical cross section) are presented for the new experiment.

*Specific Comments  Technical Corrections:*

1. *Lines 63-5: This sentence is confusing as written. I suggest deleting "as opposed to" and breaking the sentence into two sentences.*   Thank you, this now reads: "This timescale is specific to the location of the dye patch, which is off the continental shelf and responding to mesoscale processes. If the dye patch was on the shelf, one would expect a shorter timescale."

2. *Can you include a map showing the difference in tracer concentration among different model runs so that users can visually see the magnitude and spatial variability of the error of the offline simulations, compared to the online simulations?*   Yes, good idea. This is now shown in Figure 2 for a variety of offline

simulations.

3. *Figure 4 contains a lot of important information, but was difficult to understand. I suggest considering removing the 'dt' from the figure. If needed, this could be included in a subplot. Also, including nhis as a 2nd y-axis instead of numbers on the plot, would be useful for orienting the reader. Finally, drawing a box around the legend would help readers more readily separate it from the rest of the text in the figure. If the dt's are kept in the figure, please include them in the legend.* We did not remove the "dt" labels because we thought it would be more confusing to try to explain which is which simulation in words. However, we followed the spirit of your suggestions by altering much of the text in the figure to be lighter in color, and the marker edges to be lighter, so that the plot is hopefully easier to look at now, with the markers themselves standing out more. Also added a box around the legend. A note about dt was added to the legend.